# Real-World Study of Patients with Metastatic Colorectal Cancer and Long-Term Response to Regorafenib in the USA

**DOI:** 10.3390/cancers17193196

**Published:** 2025-09-30

**Authors:** Richard D. Kim, Xiaoyun Pan, Yiqiao Zhang, Orsolya Lunacsek, Federica Pisa, Helene Ostojic, Marc Peeters

**Affiliations:** 1Department of Gastrointestinal Oncology, Moffitt Cancer Center, Tampa, FL 33612, USA; 2Real World Evidence Oncology, Bayer HealthCare Pharmaceuticals Inc., Whippany, NJ 07981, USA; xiaoyun.pan@bayer.com (X.P.); yiqiao.zhang@bayer.com (Y.Z.); orsolya.lunacsek@bayer.com (O.L.); 3Real World Evidence Oncology, Bayer AG, 13353 Berlin, Germany; federica.pisa@bayer.com; 4Bayer Consumer Care AG, 4052 Basel, Switzerland; helene.ostojic@bayer.com; 5Center for Oncological Research, Antwerp University Hospital, 2650 Edegem, Belgium; ceo.marc.peeters@uza.be

**Keywords:** colorectal cancer, neoplasm metastases, response, duration of treatment

## Abstract

**Simple Summary:**

The research conducted in this study looked at treatment with regorafenib—a drug given to patients with advanced colon cancer that has spread to other parts of the body. The main objective was to determine which type of patients benefit from this drug for a long time, including demographics such as age and ethnicity, health status, and certain biological markers in their cancer cells. The study found some patients took regorafenib for at least four, five, or six months, and these patients had common characteristics, such as their health status, the side of the body where the colon cancer started, and whether disease had spread to the liver. Most had also been previously treated with the drug bevacizumab. The findings showed that patients can experience long-term benefit from regorafenib, and this information could help doctors predict who may have a better chance of responding well to this drug.

**Abstract:**

Background: Several narrative reports document long-term responses to regorafenib treatment in patients with metastatic colorectal cancer (mCRC). However, no large-scale study has assessed long-term responses and there are no established predictors of potential long-term benefit. We carried out an observational study of characteristics of patients treated in real-world clinical practice in the USA using duration of treatment (DoT) as a surrogate for treatment response. Patients and Methods: This retrospective cohort study used a de-identified electronic health record-derived database and included patients aged ≥18 years with mCRC who initiated regorafenib monotherapy between 1 July 2013 and 30 June 2023. Patient cohorts were defined by DoT ≥4 months (LTR4), ≥5 months (LTR5), or ≥6 months (LTR6) and are not mutually exclusive. Results: Of 2444 patients who initiated regorafenib monotherapy during the study, those with long-term response were analyzed: 544 had LTR4 (22%), 367 had LTR5 (15%), and 250 had LTR6 (10%). Most patients with long-term responses had left-sided tumors (65–70%), Eastern Cooperative Oncology Group performance status of 0/1 (67–68%), and liver metastases (55–61%) and had received prior bevacizumab treatment (60–67%). The median age in each group was 66 years, and patients most frequently initiated regorafenib as third-line treatment (31–33%). Median time to regorafenib discontinuation was 6.0–9.3 months among long-term responders. Conclusions: Most patients with long-term responses to regorafenib had favorable performance status at treatment initiation, left-sided tumors, and liver metastases and had received prior bevacizumab treatment. The study highlights that patients in the real-world setting were able to tolerate and maintain long-term responses to regorafenib treatment.

## 1. Introduction

Colorectal cancer (CRC) is the fourth most common cancer diagnosed in the USA, with an estimated 152,810 new cases (7.6% of all new cancer cases) and 53,010 estimated deaths (8.7% of all cancer-related deaths) in 2024 [1]. CRC remains a global public health challenge, with projections for 2040 indicating that the number of new CRC cases in the USA will reach 210,000 [2]. The 5-year relative survival rate of CRC in the USA is 91% when localized and 74% when regional but only 16% when metastatic [1]. Current standard-of-care therapies for metastatic CRC (mCRC) include fluoropyrimidine-based chemotherapy and anti-vascular endothelial growth factor or anti-epidermal growth factor receptor treatments for patients with *RAS* wild-type disease [3], whereas immunotherapies are available for patients with microsatellite instability (MSI)-high or mismatch repair (MMR)-deficient disease [3].

Regorafenib is an oral multikinase inhibitor indicated for the treatment of patients with mCRC who have been previously treated with fluoropyrimidine-, oxaliplatin-, or irinotecan-based chemotherapy, anti-vascular endothelial growth factor therapy, or, if *RAS* wild-type, anti-epidermal growth factor receptor therapy; patients with a locally advanced, unresectable, or metastatic gastrointestinal stromal tumor who have been previously treated with imatinib mesylate and sunitinib malate; and patients with hepatocellular carcinoma who have been previously treated with sorafenib at a dosage of 160 mg/day for the first 21 days of each 28-day cycle (standard dosing) [4]. Regorafenib was approved in 2013 for the treatment of mCRC based on the results of the pivotal randomized, double-blind, placebo-controlled phase III CORRECT trial (NCT01103323), which showed an overall survival benefit with regorafenib versus placebo when combined with best supportive care [5].

Of note, in CORRECT and other phase III clinical trials, 19–34% of patients with mCRC treated with regorafenib had progression-free survival (PFS) of >4 months [6,7,8]. In addition, several case reports have documented pretreated (second- to fifth-line) patients with mCRC who had long-term responses to regorafenib and were treated for 12–24 months, including an elderly patient and patients with *BRAF* or *KRAS* mutations [9,10,11,12]. Moreover, cases have been reported documenting exceptionally long PFS of ≥24 months with regorafenib treatment [13,14], and in one heavily pretreated patient receiving regorafenib, a long-term response of >9 years was reported along with good tolerability [15]. A systematic literature review demonstrated growing evidence of long-term responses with regorafenib monotherapy [16], but comprehensive real-world studies are lacking, and currently there are no established biomarkers for predicting long-term treatment benefit in patients with mCRC.

We report the results of a large-scale, retrospective, real-world study to describe the proportion, demographic, and clinical characteristics of patients with mCRC who had long-term responses to regorafenib based on duration of treatment (DoT) as a surrogate for treatment response in routine clinical practice in the USA.

## 2. Materials and Methods

### 2.1. Study Design and Patient Population

This analysis was a large-scale, retrospective, observational, real-world cohort study using data from the nationwide Flatiron Health database; this is a retrospective longitudinal electronic health record (EHR)-derived database comprising a probabilistic sample of de-identified patients diagnosed with mCRC. The clinical EHRs are derived from patient-level structured and unstructured data for approximately 37,000 patients between 2013 and 2023 and are part of a larger real-world database originating from approximately 280 oncology community practices and academic research hospitals in the USA comprising over 800 sites of care.

Patients diagnosed with mCRC from 1 January 2013 to 30 June 2023 comprised the source population. Patients had to have a diagnosis of colon or rectal cancer (International Classification of Diseases, Ninth or Tenth Revision codes 153.x, 154.x, C18x, C19x, C20x, or C21x), pathology consistent with CRC, ≥2 clinic visits in the Flatiron Health network that occurred on or after 1 January 2013, and clinical documentation of stage IV CRC or mCRC. Patients lacking relevant unstructured documents in the database for abstraction were excluded. Patients aged ≥18 years at mCRC diagnosis who initiated regorafenib between 1 July 2013 and 30 June 2023 were eligible for inclusion in this study (Figure 1). The index date was the start date of regorafenib monotherapy. Patients who had a diagnosis or recorded history of gastrointestinal stromal tumor, hepatocellular carcinoma, or other primary cancers (except non-melanoma skin cancers) during the 6-month period prior to and including the index date were excluded. Patients were required to have ≥1 recorded visit, laboratory test, or other recorded EHR entry ≥3 months prior to the index date, unless the patient was diagnosed with mCRC within 3 months prior to the index date. The follow-up period was from the first day post-index up to 30 November 2023; for each patient, end of follow-up was either the date of death or the date of the last visit of any type on or up to 30 November 2023 for patients with no record of death. This period was used for the analysis of regorafenib monotherapy duration and in the categorization of patients with mCRC as long-term responders.

The overall aim of the study was to estimate the proportion of long-term responses in patients receiving regorafenib monotherapy in real-world clinical practice and to describe the demographic and clinical characteristics of these patients. DoT with regorafenib was used as a surrogate measure of long-term response. Patients who had DoT with regorafenib of ≥4 months, ≥5 months, and ≥6 months were considered as having a long-term response of ≥4 months (LTR4), ≥5 months (LTR5), and ≥6 months (LTR6), respectively. The choice of long-term response endpoints of 4 to ≥6 months was based on medical expert opinion and clinical trial evidence of the typical proportions of patients with PFS exceeding 4 months (19–34% [6,7,8]), with median PFS ranging from 1.9 to 3.2 months [5,8,17].

### 2.2. Variables and Endpoints

Baseline patient demographic variables were age at index date, gender, race, US region, practice type, prior treatment with bevacizumab and/or fluorouracil, and index year. Disease and clinical characteristic variables were Eastern Cooperative Oncology Group (ECOG) performance status (PS) at index date, stage at initial diagnosis, serum lactate dehydrogenase (LDH) and carcinoembryonic antigen levels at index date, sidedness of primary tumor at diagnosis, sites of metastasis, number of treatment lines prior to index date, *KRAS* and *BRAF* mutation status at index date, and MMR/MSI status at index date. For LDH testing, abnormal and normal LDH values were defined based on individual patient testing and corresponding laboratory reference ranges; an LDH value was considered abnormal if it exceeded the upper limit of the reference range used by the individual laboratory and considered normal if otherwise.

The DoT was defined as the time from the start date of regorafenib treatment to its end date, as recorded in the database. For time to treatment discontinuation, patients were considered discontinuers if they (1) advanced to a new line of treatment after regorafenib; (2) did not advance to a new line of treatment but had a recorded date of death; or (3) did not advance to a new line of treatment, had no recorded date of death, and had confirmed structured activity/a visit ≥120 days after the end of regorafenib treatment. Patients who were still receiving regorafenib treatment as of their last visit date in the database were censored.

### 2.3. Statistical Analyses

A descriptive analysis was conducted for all baseline demographic, clinical characteristics, treatment characteristics, and outcomes. Standard summary statistics were provided using appropriate measures of central tendency, based on continuous or categorical data type. For continuous measures, the mean, standard deviation, median, and interquartile range were provided; for categorical measures, the number and percentage of patients were presented. Time to treatment discontinuation (in months) was assessed using the Kaplan–Meier method, and the median time with 95% confidence interval (CI) was reported. Analyses were conducted using SAS v9.4.

### 2.4. Ethics Approval and Consent to Participate

This analysis was a secondary data analysis based on patient healthcare information from the de-identified Flatiron Health database from which patients could not be de-anonymized. As dictated by Title 45 Code of Federal Regulations (45 CFR 46.101(b)(4)) (available at https://www.govinfo.gov/content/pkg/CFR-2011-title45-vol1/pdf/CFR-2011-title45-vol1.pdf (accessed on 10 December 2024)), this analysis was conducted under an exemption from Institutional Review Board oversight for US-based studies using de-identified healthcare records, and for which informed consent was not applicable.

## 3. Results

### 3.1. Patient Demographics and Clinical Characteristics

In total, 37,798 patients in the database were diagnosed with mCRC between 1 January 2013 and 30 June 2023, and, of these, 2444 patients were eligible to be included in the study (Figure 2). In the study group, 22% had LTR4 (*n* = 544), 15% had LTR5 (*n* = 367), and 10% had LTR6 (*n* = 250) (Table 1); LTR groups are not mutually exclusive (i.e., total LTR population *n* = 544). Of the 1061 patients who initiated regorafenib treatment during 2013–2018, 21% had LTR4 (*n* = 219), 14% had LTR5 (*n* = 148), and 9% had LTR6 (*n* = 98); of the 1383 patients who initiated regorafenib treatment during 2019–2022, 23% had LTR4 (*n* = 325), 16% had LTR5 (*n* = 219), and 11% had LTR6 (*n* = 152).

The demographic and clinical characteristics of patients with LTR4, LTR5, or LTR6 were similar (Table 1). Median age was 66 years (interquartile range, 58–74 years), and most patients were treated in a community practice setting (85–88% across LTR groups). Most patients with long-term responses had an ECOG performance status of 0/1 at index date (67–68%), 42–48% had stage IV advanced disease at initial mCRC diagnosis, and most had received bevacizumab and/or fluorouracil prior to regorafenib treatment initiation (60–67% and 76–79%, respectively). Of patients with long-term responses, 55–61% had liver metastases (with or without other sites of metastases) and 38–44% had non-liver metastases only. Left-sided primary tumors were more common than right-sided primary tumors (65–70% vs. 25–28%, respectively) in patients with long-term responses (Table 1).

### 3.2. Biomarkers

Among those patients tested (482/2444), serum LDH level biomarker status at index date was similar across patients with long-term responses (Table 2). Of 106 long-term responders with measured LDH levels available, 44/106 patients (42%) had serum LDH levels within the abnormal reference range at index date (range across LTR groups, 33–42%), whereas 62/106 (58%) had serum LDH levels within the normal range (range across LTR groups, 58–67%; Table 2). Median carcinoembryonic antigen level was similar across patients with long-term responses (29–36 ng/mL; Table 2). *BRAF* mutation-positive status (4%), *KRAS* mutation-positive status (22–25%), and MMR/MSI status (67–70% microsatellite stable) were similar across LTR groups (Table 2).

### 3.3. Treatment Duration and History

Overall, a higher proportion of patients with long-term responses received their regorafenib as a third-line treatment (31–33%) on the index date compared with other lines of treatment (second line, 19–22%; fourth line, 24–25%; other lines (combined), 22–24%; Figure 3). Median follow-up time from index date ranged from 10.7 months to 14.6 months (Table 3). Median time to discontinuation of regorafenib treatment was 6.0 months (95% CI: 5.7–6.3) in patients with LTR4, 7.4 months (95% CI: 7.0–7.9) in patients with LTR5, and 9.3 months (95% CI: 8.6–10.1) in patients with LTR6 (Table 3).

## 4. Discussion

This was the first large-scale, retrospective, real-world study to describe the demographic and clinical characteristics of patients with mCRC who had long-term responses to regorafenib treatment in the USA. In this study, DoT (≥4, ≥5, or ≥6 months) was used as a surrogate for long-term response, and based on this measure, over one-fifth of patients had a long-term response to regorafenib treatment of ≥4 months. Patients with long-term responses to regorafenib had similar demographic and clinical characteristics, including a favorable ECOG PS at regorafenib treatment initiation. Most long-term responders had left-sided tumors, highlighting the potential prognostic value of tumor sidedness; this has been noted in previous studies showing left-sided tumors were associated with a positive effect on disease control rate following regorafenib treatment [18]. Moreover, long-term responses were seen despite the presence of liver metastases (with or without other sites of metastases) in the majority of patients.

There is currently no established standardized predictive approach for identifying patients with mCRC who are most likely to derive prolonged benefit from treatment with regorafenib monotherapy. A low level of serum LDH and *HERG1* and *EPAS1* gene protein expression have been identified as potential predictors of response to anti-angiogenic therapies [19,20]. In addition, good ECOG PS, presence of lung-limited metastatic disease, growth arrest specific 6 (*GAS6*) gene amplification, and SMAD family member 4 (*SMAD4*) mutation have been identified as potential factors predictive of better prolonged response to regorafenib [21]. In our study, serum LDH level biomarker status at index date was similar across patients with long-term responses; the majority of patients with LDH data available and with long-term responses had serum levels within the normal reference range at index date, although overall numbers tested were small. Patients with LTR5 or LTR6 tended to have lower median carcinoembryonic antigen levels than those with LTR4. In addition, a slightly greater proportion of patients with LTR5 or LTR6 had left-sided tumors and non-liver metastases only, compared with those with LTR4. However, identification of predictive factors was not a study objective, and statistical analyses were not carried out. Comprehensive molecular characterization data were unavailable in this study, but among patients tested, *BRAF* and *KRAS* mutation-positive status and MMR/MSI status were similar across long-term responder groups, including when the LTR cut-off time was expanded from ≥4 to ≥6 months.

Most patients with long-term responses had received prior bevacizumab treatment despite a prior study in Asian patients suggesting that exposure to prior targeted treatments, such as bevacizumab, may lead to a relatively shorter survival benefit with regorafenib treatment compared with the global population [17]. A smaller proportion of patients with LTR5 or LTR6 received prior bevacizumab treatment compared with those with LTR4 (as noted above, statistical analyses were not carried out). The proportion of patients with long-term responses who initiated regorafenib treatment in 2019 or later was slightly higher than the proportion with long-term responses who initiated treatment before 2019. This small increase coincided with completion of the phase II ReDOS study that evaluated alternative approaches for optimizing regorafenib dosing in clinical practice [22], and this may have led to improved dosing schedules and tolerability thereafter. These results align with previous studies that have used other long-term response definitions, such as PFS >4 months, and that have shown that less advanced disease at initial diagnosis and a favorable performance status are associated with long-term responses to regorafenib treatment [6,8,23,24].

### 4.1. Strengths

The strengths of this study include that it estimates the prevalence of patients who experience long-term responses to regorafenib treatment in real-world clinical practice and provides the demographics, clinical characteristics, and treatment history of those patients using a large and representative EHR database sourced primarily from community oncology clinics in the USA. Although external generalizability may be considered limited because the collected real-world data originate largely from community oncology settings, this setting is nonetheless considered as representative of typical care in the USA since 80–85% of patients with cancer are treated in community settings, particularly in rural areas [25].

### 4.2. Limitations

As with other retrospective database studies, this study has some limitations that mean results should be interpreted with caution. Limitations of EHR database studies include gaps in information on treatments received, outcomes, and events that occur outside the documented network database of oncology practices, or when a patient is lost to follow-up. Underreporting or misclassification of disease history, or other demographic and clinical characteristics that may help characterize the patient’s actual health status, is also possible. This study is also limited by its use of a proxy measure of response to treatment rather than documented response to treatment or progression recorded in the database. Reasons for discontinuation are unknown and may include poor tolerability rather than lack of response. Furthermore, censored patients may not have been included in all relevant LTR groups due to their full DoT being unavailable.

A further limitation is that the current study population may not be representative of all patients with mCRC treated in the USA as 45% of the study cohort were from the Southern region of the USA and 62% were White. These factors may limit the generalizability of the findings to other regions of the USA and racial groups, respectively, due to potential differences in prevalences of comorbidities and unfavorable tumor characteristics in addition to variable treatment access by race and/or according to rural versus urban regions [26,27].

## 5. Conclusions

Overall, approximately 22% of the total eligible patient population in this study received long-term regorafenib treatment of ≥4 months. Patients who derived the longest benefit from regorafenib treatment had similar demographic and clinical characteristics, including a favorable ECOG performance status at regorafenib initiation; most had left-sided tumors, and liver metastases and had received prior bevacizumab treatment. The overall findings highlight that patients in the real-world setting were able to tolerate and maintain long-term responses to regorafenib treatment.

## Figures and Tables

**Figure 1 cancers-17-03196-f001:**
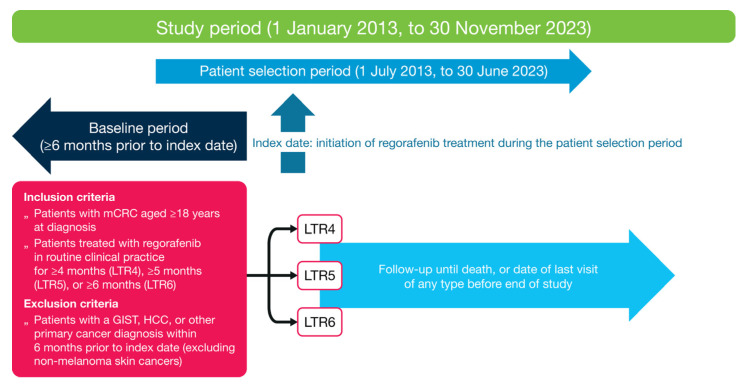
Study design. GIST, gastrointestinal stromal tumor; HCC, hepatocellular carcinoma; LTR, long-term response; mCRC, metastatic colorectal cancer.

**Figure 2 cancers-17-03196-f002:**
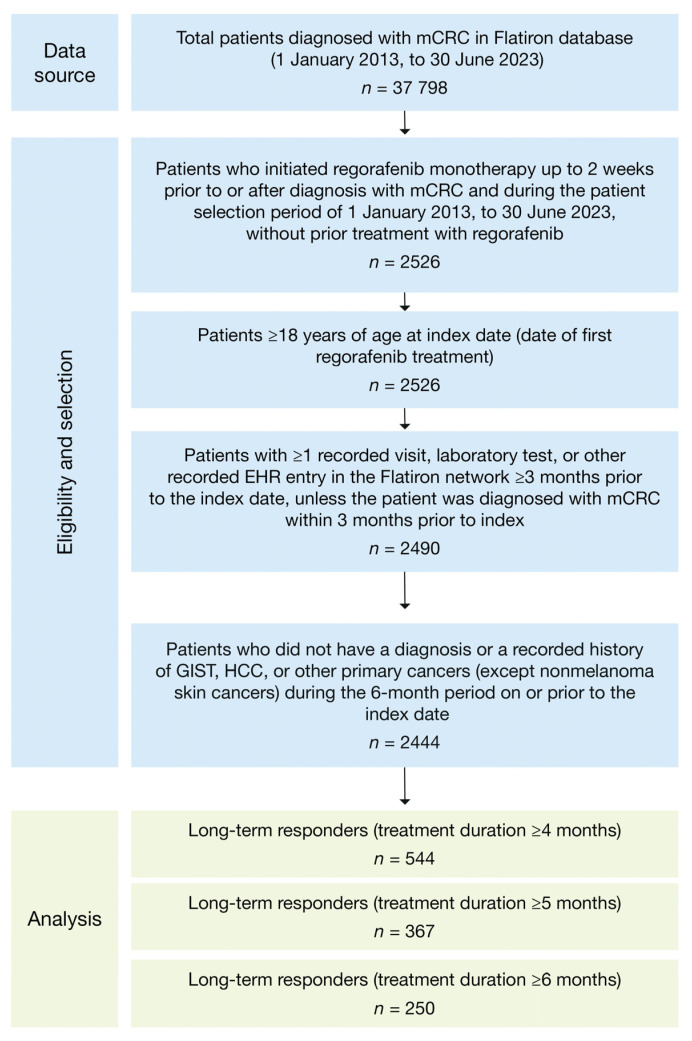
Attrition flowchart. EHR, electronic health record; GIST, gastrointestinal stromal tumor; HCC, hepatocellular carcinoma; mCRC, metastatic colorectal cancer.

**Figure 3 cancers-17-03196-f003:**
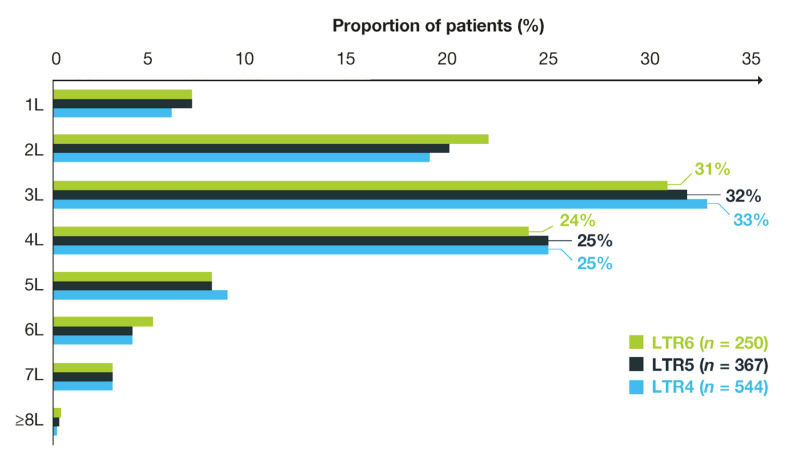
Long-term responders to regorafenib by treatment line. L, line of treatment; LTR, long-term response.

**Table 1 cancers-17-03196-t001:** Patient demographic and clinical characteristics at index date.

Characteristics	LTR4 ^a^ (*n* = 544)	LTR5 ^a^ (*n* = 367)	LTR6 ^a^ (*n* = 250)
**Male gender**, *n* (%)	309 (57)	207 (56)	141 (56)
**Age** Median (IQR), years ≤65, *n* (%) >65, *n* (%)	66 (58–74) 271 (50) 273 (50)	66 (58–74) 183 (50) 184 (50)	66 (58–74) 124 (50) 126 (50)
**Race**, *n* (%) White Black or African American Asian Other Unknown	349 (64) 73 (13) 15 (3) 81 (15) 26 (5)	243 (66) 47 (13) 10 (3) 51 (14) 16 (4)	167 (67) 30 (12) 4 (2) 38 (15) 11 (4)
**Region**, *n* (%) Midwest Northeast South West Unknown	59 (11) 84 (15) 244 (45) 80 (15) 77 (14)	39 (11) 60 (16) 158 (43) 55 (15) 55 (15)	21 (8) 37 (15) 111 (44) 39 (16) 42 (17)
**Stage at initial CRC diagnosis**, *n* (%) 0/I II III IV Unknown	16 (3) 85 (16) 164 (30) 261 (48) 18 (3)	12 (3) 53 (14) 119 (32) 170 (46) 13 (4)	11 (4) 43 (17) 83 (33) 104 (42) 9 (4)
**ECOG PS**, *n* (%) 0/1 2–4 Unknown/missing	366 (67) 64 (12) 114 (21)	251 (68) 48 (13) 68 (19)	170 (68) 31 (12) 49 (20)
**Sidedness of tumor**, *n* (%) ^b^ Left Right Other/missing	356 (65) 152 (28) 36 (7)	249 (68) 95 (26) 23 (6)	176 (70) 62 (25) 12 (5)
**Sites of metastases**, *n* (%) ^c^ Liver ± other sites Non-liver only Missing	333 (61) 209 (38) 2 (<1)	212 (58) 153 (42) 2 (<1)	138 (55) 110 (44) 2 (1)
**Practice type**, *n* (%) Academic Community Academic and community	53 (10) 476 (88) 15 (3)	35 (10) 319 (87) 13 (4)	27 (11) 213 (85) 10 (4)
**Received prior bevacizumab treatment**, *n* (%)	367 (67)	231 (63)	149 (60)
**Received prior fluorouracil treatment**, *n* (%)	431 (79)	280 (76)	190 (76)
**Initiated regorafenib treatment between 2013 and 2018**, *n* (%)	219 (40)	148 (40)	98 (39)
**Initiated regorafenib treatment between 2019 and 2022**, *n* (%)	325 (60)	219 (60)	152 (61)

Abbreviations: CRC, colorectal cancer; ECOG PS, Eastern Cooperative Oncology Group performance status; IQR, interquartile range; LTR, long-term response. ^a^ LTR groups are not mutually exclusive (i.e., total LTR population *n* = 544); ^b^ Left side includes rectosigmoid, rectum, and splenic flexure. Patients with multiple records of sidedness were categorized as other/missing; ^c^ Metastasis occurred at any time before or on index date.

**Table 2 cancers-17-03196-t002:** Biomarker and laboratory test status at index date in patients with an LTR to regorafenib treatment.

Biomarker/ Laboratory Test	LTR4 (*n* = 544)	LTR5 (*n* = 367)	LTR6 (*n* = 250)
**LDH levels**, *n* (%) Low: <300 IU/L High: ≥300 IU/L Unknown/missing Within normal reference range ^a^ Within abnormal reference range ^a^ Unknown/missing	74 (14) 34 (6) 436 (80) 62 (11) 44 (8) 438 (81)	56 (15) 20 (5) 291 (79) 49 (13) 26 (7) 292 (80)	36 (14) 14 (6) 200 (80) 33 (13) 16 (6) 201 (80)
**Median CEA levels** (IQR), ng/mL ^b^	36 (8–152)	32 (8–139)	29 (7–108)
*BRAF ***mutation status**, *n* (%) Positive Negative Unknown/missing	20 (4) 335 (62) 189 (35)	13 (4) 223 (61) 131 (36)	10 (4) 146 (58) 94 (38)
*KRAS ***mutation status**, *n* (%) Positive Negative Unknown/missing	134 (25) 107 (20) 303 (56)	89 (24) 80 (22) 198 (54)	55 (22) 63 (25) 132 (53)
**MMR/MSI status**, *n* (%) MSS dMMR/MSI-H Unknown/missing	379 (70) 9 (2) 156 (29)	253 (69) 6 (2) 108 (29)	168 (67) 6 (2) 76 (30)

Abbreviations: CEA, carcinoembryonic antigen; dMMR, mismatch repair deficiency; IQR, interquartile range; LDH, lactate dehydrogenase; LTR, long-term response; MMR, mismatch repair; MSI, microsatellite instability; MSI-H, high microsatellite instability; MSS, microsatellite stable. ^a^ Abnormal and normal LDH values were defined based on individual patient testing and corresponding laboratory reference ranges; an LDH value was considered abnormal if it exceeded the upper limit of the reference range used by the individual laboratory, and normal otherwise; ^b^ Data available for 410 patients with LTR4, 280 patients with LTR5, and 195 patients with LTR6.

**Table 3 cancers-17-03196-t003:** Treatment duration and history of patients with an LTR to regorafenib.

Treatment Duration/ History Variable	LTR4 (*n* = 544)	LTR5 (*n* = 367)	LTR6 (*n* = 250)
**Median time to discontinuation of regorafenib treatment** (95% CI), ^a^ months	6.0 (5.7–6.3)	7.4 (7.0–7.9)	9.3 (8.6–10.1)
**Median follow-up time from index date** (IQR), months	10.7 (7.2–17.2)	13.0 (8.1–20.2)	14.6 (9.9–22.3)

Abbreviations: CI, confidence interval; IQR, interquartile range; LTR, long-term response.^a^ Analyzed using the Kaplan–Meier method.

## Data Availability

The data that support the findings of this study were collected by and are the property of Flatiron Health, Inc., which has restrictions prohibiting the authors from making the data set publicly available. Requests for data sharing by license or by permission for the specific purpose of replicating results in this manuscript can be submitted to PublicationsDataAccess@flatiron.com (accessed on 10 December 2024).

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
