# Peer review of "Real-World Study of Patients with Metastatic Colorectal Cancer and Long-Term Response to Regorafenib in the USA"

_cancers, 2025, doi:10.3390/cancers17193196_

Round 1

Reviewer 1 Report

Comments and Suggestions for Authors

While the topic is of potential interest, I find that a control group would be necessary to identify predictive factors of benefit from regorafenib treatment. The study presented is purely descriptive without any statistical calculations.

Author Response

Comment 1: While the topic is of potential interest, I find that a control group would be necessary to identify predictive factors of benefit from regorafenib treatment. The study presented is purely descriptive without any statistical calculations.

Response 1: [Lines 262‒267 and Lines 274‒276] 

The objective of our real-world study was to describe the proportion, demographics, and clinical characteristics of patients with metastatic colorectal cancer (mCRC) who had long-term response (LTR) to regorafenib based on duration of treatment (DoT) as a surrogate for LTR set at three different thresholds. The characteristics we describe may serve as a foundation for future research into potential predictive factors of treatment benefit, but predictive modeling was not our aim. We agree with the reviewer that future research should focus on finding statistically significant and clinically relevant drivers of benefit based on appropriately controlled study design and inferential methodology.  

Although it was not a study objective to identify predictive factors of benefit from regorafenib treatment, we have now added detail of apparent differences across cohorts LTR4, LTR5, and LTR6 in the Discussion, while noting that this was not a study aim and statistical analyses were not carried out. 

Reviewer 2 Report

Comments and Suggestions for Authors
  • In Table 1, it would be valuable to consider including additional stratification characteristics such as prior treatment history, comorbidities, socioeconomic factors, and lifestyle behaviors (e.g., smoking and alcohol consumption). Incorporating these multidimensional features could enable a more comprehensive analysis of factors.
  • The manuscript mentions the use of Duration of Treatment (DoT ≥4, ≥5, or ≥6 months) as a surrogate marker for long-term response (LTR). However, it is unclear which specific threshold was ultimately applied in the primary analyses. Clarifying this is important, as it directly impacts how patient groups were defined and interpreted throughout the study.

  • The authors state that “this setting is nonetheless considered as representative of the care setting in which patients with cancer are typically treated in the USA.” However, no clear justification or supporting evidence is provided for this claim. It would strengthen the argument.

  • In Section 4.2, the authors acknowledge that the demographic composition of the cohort may limit the generalizability of the findings. A more in-depth discussion of this issue would be helpful to contextualize the findings and guide future research.

Author Response

Comment 1: In Table 1, it would be valuable to consider including additional stratification characteristics such as prior treatment history, comorbidities, socioeconomic factors, and lifestyle behaviors (e.g., smoking and alcohol consumption). Incorporating these multidimensional features could enable a more comprehensive analysis of factors.

Response 1: [Table 1, Line 132 and Lines 198‒199] 

Although limited, data on prior treatment, including bevacizumab, were available and are included in Table 1. Data on prior fluorouracil treatment were also available and have been added. 

The electronic health record (EHR)-derived database used for this study did not contain variables related to socioeconomic factors or lifestyle behaviors. Comorbidities were only available as diagnosis codes recorded in the structured data and were not abstracted from progress notes. We found that comorbidities were underreported and presumably incomplete in this oncology EHR database, which is consistent with prior research on the topic. We agree with the reviewer that future research should combine EHR data with other data sources to enhance the completeness of the list of potential predictive factors. 

Comment 2: The manuscript mentions the use of Duration of Treatment (DoT ≥4, ≥5, or ≥6 months) as a surrogate marker for long-term response (LTR). However, it is unclear which specific threshold was ultimately applied in the primary analyses. Clarifying this is important, as it directly impacts how patient groups were defined and interpreted throughout the study.

Response 2: [Lines 172‒173] The protocol-defined primary objective was to determine the proportion of patients with mCRC who received treatment with regorafenib monotherapy for ≥5 months. Nonetheless, all three LTR thresholds were ultimately used in the final analysis as described in Section 2.1; however, the LTR groups are not mutually exclusive, and the total LTR population comprised 544 patients. This has been stated in a footnote to Table 1 and has also now been clarified in the text.  

Comment 3: The authors state that “this setting is nonetheless considered as representative of the care setting in which patients with cancer are typically treated in the USA.” However, no clear justification or supporting evidence is provided for this claim. It would strengthen the argument.

Response 3: [Lines 294-295. Reference 25. Line 422‒423] Approximately 8085% of patients with cancer in the USA are treated in community settings (higher in rural areas), and this is reflected in the Flatiron database, in which 8588% of patients were from community practice across the LTR groups. The supporting evidence has now been included in the Discussion, together with a supporting reference (Tucker, T.C.; et al. Ann Surg Oncol 2021;28:632638).

Comment 4: In Section 4.2, the authors acknowledge that the demographic composition of the cohort may limit the generalizability of the findings. A more in-depth discussion of this issue would be helpful to contextualize the findings and guide future research.

Response 4: [Lines 310‒314] The discussion of potential demographic factors that may affect the generalizability of findings has been expanded to include some potential specific factors that may vary by region and/or race.

Reviewer 3 Report

Comments and Suggestions for Authors

For the patients in this study, is there any genome-level similarity or differences?

Author Response

Comment 1: For the patients in this study, is there any genome-level similarity or differences?

Response 1: With the exception of BRAF and KRAS mutation (included in Table 2), genome-level data were not available/collected for this study from the Flatiron database.

Reviewer 4 Report

Comments and Suggestions for Authors

Thank you for giving me the valuable opportunity to review this manuscript. I have several comments.
1) The manuscript presents a comparison of patient backgrounds among LTR4, LTR5 and LTR6. Have statistical significance tests been performed on these comparisons? If so, please include the results.
2) In major phase III trials, regorafenib has been reported to yield a progression-free survival (PFS) of 1.9–3.2 months. Primary interest lies in identifying differences between patients with a duration of response of less than four months and those with a duration of response of four months or more (i.e. LTR4, LTR5 and LTR6). I believe this information should be presented.
3) In the present study, the proportion of patients with BRAF mutations is approximately 60%, which is notably high, while the proportion with KRAS mutations is approximately 20%, which is relatively low. I suggest that the potential reasons for this distribution are addressed in the Discussion section.

Author Response

Comment 1: The manuscript presents a comparison of patient backgrounds among LTR4, LTR5 and LTR6. Have statistical significance tests been performed on these comparisons? If so, please include the results.

Comment 2: In major phase III trials, regorafenib has been reported to yield a progression-free survival (PFS) of 1.9–3.2 months. Primary interest lies in identifying differences between patients with a duration of response of less than four months and those with a duration of response of four months or more (i.e. LTR4, LTR5 and LTR6). I believe this information should be presented. 

Response 1 and Response 2: [Lines 262‒267 and Lines 274‒276] 

Our objective was to quantify LTR using three progressively stricter definitions (≥4 months’, ≥5 months’, and ≥6 months’ DoT with regorafenib), and to describe patients who reached LTR based on these definitions. We did not plan to compare the patients who met these thresholds, but we agree with the reviewer that this investigation would be a valuable and logical extension of our research.  

We have added preliminary discussion of potential trends that might require further study, while noting that this was not an objective of the study and statistical tests were not carried out.

Comment 3: In the present study, the proportion of patients with BRAF mutations is approximately 60%, which is notably high, while the proportion with KRAS mutations is approximately 20%, which is relatively low. I suggest that the potential reasons for this distribution are addressed in the Discussion section.

Response 3: BRAF mutation-positive status was found in 4% of patients, which is lower than the 8‒12% expected, partly due to missing information. Similarly, KRAS mutation-positive status (~25%) was lower than expected, which is likely an artifact of the large proportion of patients (>50%) who had missing KRAS information. If we remove patients with missing KRAS information from the denominator, KRAS-positive status is found in >50% of patients with known KRAS status. Since there were no apparent differences across LTR groups, these data were not discussed further.